# The Kinetics of Lymphatic Dysfunction and Leukocyte Expansion in the Draining Lymph Node during LTB_4_ Antagonism in a Mouse Model of Lymphedema

**DOI:** 10.3390/ijms22094455

**Published:** 2021-04-24

**Authors:** Matthew T. Cribb, Lauren F. Sestito, Stanley G. Rockson, Mark R. Nicolls, Susan N. Thomas, J. Brandon Dixon

**Affiliations:** 1Parker H. Petit Institute for Bioengineering and Bioscience, Georgia Institute of Technology, Atlanta, GA 30332, USA; mcribb3@gatech.edu (M.T.C.); susan.thomas@gatech.edu (S.N.T.); 2George W. Woodruff School of Mechanical Engineering, Georgia Institute of Technology, Atlanta, GA 30332, USA; 3Wallace H. Coulter Department of Biomedical Engineering, Georgia Institute of Technology, Atlanta, GA 30332, USA; lfsestito@gatech.edu; 4Stanford University School of Medicine, Stanford University, Stanford, CA 94305, USA; rockson@stanford.edu (S.G.R.); mnicolls@stanford.edu (M.R.N.); 5VA Palo Alto Health Care System, Palo Alto, CA 94304, USA

**Keywords:** lymphedema, leukocyte, lymphatic, near-infrared imaging, lymph node, leukotriene B4, bestatin, immune response

## Abstract

The mechanisms of lymphedema development are not well understood, but emerging evidence highlights the crucial role the immune system plays in driving its progression. It is well known that lymphatic function deteriorates as lymphedema progresses; however, the connection between this progressive loss of function and the immune-driven changes that characterize the disease has not been well established. In this study, we assess changes in leukocyte populations in lymph nodes within the lymphatic drainage basin of the tissue injury site (draining lymph nodes, dLNs) using a mouse tail model of lymphedema in which a pair of draining collecting vessels are left intact. We additionally quantify lymphatic pump function using established near infrared (NIR) lymphatic imaging methods and lymph-draining nanoparticles (NPs) synthesized and employed by our team for lymphatic tissue drug delivery applications to measure lymphatic transport to and resulting NP accumulation within dLNs associated with swelling following surgery. When applied to assess the effects of the anti-inflammatory drug bestatin, which has been previously shown to be a possible treatment for lymphedema, we find lymph-draining NP accumulation within dLNs and lymphatic function to increase as lymphedema progresses, but no significant effect on leukocyte populations in dLNs or tail swelling. These results suggest that ameliorating this loss of lymphatic function is not sufficient to reverse swelling in this surgically induced disease model that better recapitulates the extent of lymphatic injury seen in human lymphedema. It also suggests that loss of lymphatic function during lymphedema may be driven by immune-mediated mechanisms coordinated in dLNs. Our work indicates that addressing both lymphatic vessel dysfunction and immune cell expansion within dLNs may be required to prevent or reverse lymphedema when partial lymphatic function is sustained.

## 1. Introduction

Lymphedema is a debilitating disease that affects millions of people around the world and is characterized by tissue fibrosis, limb swelling, and recurrent soft tissue infections [1]. There are currently no pharmacological treatments for lymphedema; however, the immune response has been heavily implicated in lymphedema development [2,3,4,5,6,7]. Multiple studies have identified possible treatments for lymphedema that target the various immune-mediated changes that characterize the disease [8,9,10]. One such possible treatment is bestatin, a drug which antagonizes the production of leukotriene B4 (LTB_4_), a lipid mediator of inflammation and a metabolite of arachidonic acid [11]. A previous study found that bestatin treatment reduced swelling and improved lymphatic function in a mouse model of lymphedema through the drug’s antagonism of LTB4, which is elevated in lymphedema both in mouse models and clinically and has strong anti-lymphangiogenic effects at high concentrations [12]. The culmination of these various studies strongly suggests that both impaired lymphangiogenesis and leukocytes, including T cells and macrophages, drive tissue fibrosis and lymphatic dysfunction during lymphedema development.

Previous reports have focused primarily on studying leukocyte infiltration into lymphedematous tissue to explain their effects on the existing lymphatic vasculature and the resultant swelling. Few studies have examined how leukocyte populations in downstream lymph nodes (LNs) respond during lymphedema. Garcia Nores et al. found that CD4+ T cells were activated by dendritic cells in regional LNs and migrated to the site of injury to initiate lymphedema [13]. Leukocyte populations within LNs draining the skin of mice lacking dermal lymphatic vessels were also previously reported in genetic models of disease [14,15]. To our knowledge, no studies have investigated how multiple leukocyte populations in the draining lymph nodes (dLNs) change as swelling worsens during lymphedema progression, however, which is of key clinical importance since the onset of lymphedema is a delayed process and fluid at the site of disease onset is continuously drained to downstream LNs [16].

Many previous studies of lymphedema used the mouse tail double vessel ligation model where all initial and collecting lymphatics in the tail are blocked due to circumferential ligation of lymphatics encircling the tail [17,18]. This model results in minimal fluid and leukocyte uptake to the draining lymph nodes after the injury due to the complete blockage of lymph outflow from the lymphedema site. However, in human lymphedema, many drainage pathways from the affected limb are still intact, suggesting that drainage does not need to be fully inhibited for lymphedema to develop. An adaptation of this lymphedema model developed previously by our lab, the single vessel ligation model, leaves a pair of collecting lymphatic vessels intact while ligating the other initial and collecting lymphatic vessels. This procedure leaves an intact drainage pathway on one side of the tail while the other pathway is completely blocked, but the progression of lymphedema in this model mirrors the typical mouse tail model even though an intact drainage pathway exists [19].

This study presents a detailed analysis of time-varying changes in LN leukocyte populations during lymphedema using this single vessel ligation model. In so doing, both the immune response within the dLN in lymphedema and how flow into the LN may modulate these changes were simultaneously delineated. Moreover, we utilize this model to explore the kinetics of transport, swelling, and immune cell populations within the dLN in the context of LTB_4_ antagonism with bestatin. Our findings demonstrate that leukocyte populations extensively expand in dLNs following lymphedema induction, and this increase in lymph node leukocyte populations is correlated with the accumulation of NPs draining the injury site within the dLNs. Additionally, we show that treatment with bestatin increases lymphatic contractile transport and NP accumulation during lymphedema progression but does not reduce the observed leukocyte expansion in the dLNs or the tail swelling response characteristic of lymphedema.

## 2. Results

### 2.1. Leukocyte Populations Expand in dLNs during Lymphedema Progression

To study how various immune cell populations in dLNs change during lymphedema progression, we used a single vessel ligation lymphedema mouse tail model developed by our group [19]. This model leaves one collecting vessel trunk intact, allowing for drainage from the site of injury to the LN downstream of the functioning vessel, referred to as the intact vessel dLN, while the LN downstream of the ligated vessel, referred to as the injured vessel dLN, experiences inhibited lymph inflow. Both sacral lymph nodes (dLNs) that normally drain the mouse tail were harvested at multiple timepoints (2 days, 1, 2, and 3 weeks) following lymphedema induction, dispersed into single cell suspensions, stained with antibodies against a variety of leukocyte markers, and analyzed on a flow cytometer. We found that the sum of leukocyte populations in both dLNs significantly increased during the swelling following lymphedema induction. The number of various leukocyte populations were overall increased in summed dLNs three weeks post-surgery (3W), including T (CD3+) and B (CD19+) cells, myeloid cells (CD11b+), monocytes (CD11b+ CD64+), macrophages (CD11b+ F480+), M1 macrophages (CD11b+ F480+ MHCII+), M2 macrophages (CD11b+ F480+ CD206+), dendritic cells (CD11c+ MHCII+), and neutrophils (CD11b+ Ly6G+) (Figure 1A and Appendix A). With respect to their distribution amongst total leukocytes, T cell frequencies decreased somewhat whereas B cells increased by 3W (Figure 1B). Frequencies of dendritic cells, monocytes, macrophages, M1 macrophages, M2 macrophages, and neutrophils also significantly increased by 3W (Figure 1B and Appendix A). This data suggests that leukocytes expand extensively in the dLNs following lymphedema induction in a manner that is not specific to a particular leukocyte subset. Interestingly, the dLNs transition from a T cell majority composition to a roughly equal number of T and B cells as swelling increases. It remains to be investigated the extent that this change in distribution is driven by differences in lymphocyte proliferation or migration.

### 2.2. T Helper Cells Increase as a Percentage of T Cells in dLNs during Lymphedema Development

Previous studies have identified CD4+ T helper cells as key drivers of lymphedema progression, including their roles in increasing tissue fibrosis and reducing lymphatic function [6,13]. To analyze how T cell subpopulations in the dLNs change during disease progression in our model, we used flow cytometry to again analyze cell populations in the dLNs, this time analyzing changes in T helper cells (CD3+ CD4+) and cytotoxic T cells (CD3+ CD8+). Given the observed changes in T cells shown in Figure 1, we quantified these T cell subpopulations only at 2W following lymphedema surgery, when the increase in T cells in the dLNs was first observed. Compared to an unoperated control, both T helper cells and cytotoxic T cells were increased in dLNs at the 2W timepoint (Figure 2A). However, we found that T helper cells as a percentage of T cells increased in dLNs at the 2W timepoint, while there was no measured change in cytotoxic T cells as a percentage of T cells (Figure 2B). We also found a positive correlation between change in tail volume and percentage of T cells for T helper cells (*p =* 0.1323), while there was a negative correlation for cytotoxic T cells (*p =* 0.1150) (Figure 2C). These results show that CD4+ T helper cells make up a larger proportion of T cells in the dLNs as swelling manifests during lymphedema progression. This data corresponds well with previous studies which have shown similar increases in CD4+ T helper cells in lymphedematous tissue and have implicated CD4+ T helper cell migration from dLNs to tissue as a key driver of lymphedema [6,13].

### 2.3. Tail Swelling Correlates with Changes in Leukocyte Populations in dLNs

Similar to the varied presentation of lymphedema clinically, the single vessel ligation lymphedema mouse tail model used in this study results in a swelling response of the tail that varies amongst animals in its severity as well as rate of progression. This allows how the extent of swelling impacts leukocyte populations in the intact dLN to be studied. The swelling observed herein showed a similar increase in tail volume as observed in previous reports, with percent volume change reaching a maximum at 3W (Appendix A). Using Spearman’s rank correlation, we found that the numbers of T cells, B cells, dendritic cells, macrophages, M1 macrophages, monocytes, and neutrophils within the intact vessel dLN were all positively correlated with the percent change in volume of the tail (Figure 3A and Appendix A). This result showed that leukocyte expansion in the intact vessel dLN was related to the extent of tail swelling observed in this lymphedema model. Additionally, we found that leukocyte populations as a fraction of total leukocytes in the intact vessel dLN correlated with swelling for a variety of cell types. T cell fraction negatively correlated with swelling, while B cell fraction positively correlated with swelling (Figure 3B). Other leukocyte populations, including dendritic cells, macrophages, M1 macrophages, monocytes, and neutrophils, also had positive correlations between swelling and frequency (Figure 3B and Appendix A). These results follow from the changes observed in leukocyte populations in the dLNs over time as swelling also increased by the 3W timepoint in this model.

### 2.4. Increased Accumulation of Lymph-Draining Nanoparticles within Intact Vessel dLN Compared to Injured Vessel dLN Following Single Vessel Ligation Diminishes as Swelling Progresses

To measure transport to and NP accumulation within dLNs, we used a nanoparticle system that has been previously shown to preferentially drain into lymphatic vessels after injection and accumulate within dLNs [20,21,22,23,24]. These NPs were covalently conjugated to fluorophore IRdye 680RD (LI-COR Biotechnology, Lincoln, NE, USA) using an irreversible linker and intradermally injected into the tip of the tail one day prior to harvesting of the dLNs. After dissecting the lymph nodes at various time points, we measured the fluorescence in the nodes at the 700 nm emission wavelength of the dye using an Olympus near-infrared (NIR) imaging microscope. Using these images (Figure 4A), we calculated multiple metrics including maximum fluorescence intensity within the lymph node, mean fluorescence intensity within the lymph node (normalized to background fluorescence), and the sum of the fluorescence intensity within the lymph node (normalized to background fluorescence). The natural logarithm was used to transform each metric to form a normal distribution and allow for statistical analysis using parametric methods. NP fluorescence (both the maximum measured signal and normalized mean) within the intact vessel dLN decreased significantly at 2W compared to 2D, and at 3W compared to 2D and 1W (Figure 4B). The size of the intact vessel dLN also significantly increased between 2D and 3W, in alignment with the flow-cytometrically measured cellular expansion of the intact vessel dLN leukocyte populations (Figure 4B). The normalized sum of fluorescence in the intact vessel dLN did not significantly change at any timepoint, possibly indicating that the reduction in mean and maximum fluorescence was due to the same amount of NP spread out over the increased LN area (Figure 4B). From the metrics used in our analysis, normalized sum of fluorescence best represents overall nanoparticle drainage to the dLN given that it represents the total fluorescent signal from NPs trafficked to the dLN. It is important to note that the increased LN area did not result in increased NP accumulation as measured by normalized sum of fluorescence, even though the capacity of the lymph node for NP accumulation likely increases with LN area. The normalized sum of fluorescence increased in the injured vessel dLN at 3W compared to 2D and 1W, suggesting that partial collateralization of the lymphatic network had occurred, and transport had been partially restored to the injured vessel dLN (Figure 4C). Additionally, the ratio of NP fluorescence in the intact vessel dLN to that in the injured vessel dLN was significantly increased at 2D and 1W compared to 3W and at 2D compared to 2W for maximum fluorescence. The ratio was also significantly increased at 2D and 1W compared to 2W and 3W for normalized mean fluorescence and at 2D and 1W compared to 3W for normalized sum of fluorescence (Figure 4D). The change in this ratio shows that NP accumulation in the intact vessel dLN was significantly higher than the injured vessel dLN within 1W of the surgery, while after 2W as swelling progresses the differences between the injured and intact vessel dLNs were diminished. Restoration of transport to the injured vessel dLN partially explains the loss of differences between the dLNs as lymphedema developed.

### 2.5. Increased Leukocyte Expansion in the Intact Vessel dLN Following Single Vessel Ligation

Immune cell expansion in the intact vessel dLN was compared to that of the injured vessel dLN post injury. We found no significant differences between intact and injured vessel dLNs for any of the leukocyte populations we surveyed (Appendix A). However, certain trends were apparent, including an increase in number of cells in the intact vessel dLN compared to the injured vessel dLN at the 1W timepoint. Equation (1) shows the calculation for a metric we call normalized difference,
(1)$$Normalized\;Difference=\frac{\#\;of\;cells\;\left(Intact\;Vessel\;dLN\right)-\#\;of\;cells\;\left(Injured\;Vessel\;dLN\right)}{\#\;of\;cells\;\left(Intact\;Vessel\;dLN\right)+\#\;of\;cells\;\left(Injured\;Vessel\;dLN\right)}$$
which is positive if more cells are in the intact vessel dLN, and negative if more cells are in the injured vessel dLN. Normalized difference was significantly increased at 1W compared to 3W for monocytes, macrophages, and M1 macrophages (Figure 5A). Additionally, a similar trend at 1W was observed for all other leukocyte populations. This indicates that a larger proportion of leukocytes were within the intact vessel dLN at 1W compared to the injured vessel dLN, but these differences were lost by 2W. Interestingly these differences were most pronounced in cell populations known to be highly migratory, including monocytes and macrophages. Previous studies in our lab have shown that contractile function in the intact lymphatic vessel in this model decreases significantly by the 2W timepoint [19]. The loss of preferential proliferation and migration in the intact vessel dLN may be due to loss of function in the intact vessel [19]. To analyze whether lymphatic transport to the dLNs as measured by NIR-labeled NP accumulation within the dLNs was correlated with local leukocyte expansion, we plotted normalized difference for the total number of live cells versus the intact/injured ratio for maximum, normalized mean, and normalized sum of fluorescence. We found that normalized difference was positively correlated with each of these metrics (Figure 5B). This indicates that the differential transport of NPs to the dLNs was correlated with the differential expansion of cells within the dLNs. Normalized difference was also positively correlated with the intact/injured ratio for LN area, showing that the number of cells identified via flow cytometry correlated with the size of the dLN (Figure 5B).

### 2.6. Bestatin Treatment Increases Nanoparticle Accumulation within the Intact Vessel dLN during Later Stages of Lymphedema Progression

Leukotriene B4 (LTB4) is a known chemoattractant and has been shown to play a bimodal role in lymphangiogenesis, with low concentrations being lymphangiogenic and high concentrations being anti-lymphangiogenic [12,25,26]. Previous studies suggest that LTB4 levels increase during lymphedema, and these high levels of LTB4 in lymphedema inhibit lymphangiogenesis and resolution of swelling, thus driving progression of the disease [12]. We were interested in studying how antagonism of LTB4 may modulate lymphatic function during lymphedema progression. To approach this question, we treated mice with daily injections of bestatin, an LTB4 antagonist, starting 3 days after surgery. To determine how lymphatic pump function is altered due to bestatin treatment during lymphedema progression, we used NIR imaging techniques that have been previously described [19,27,28,29]. At 1W, 2W, and 4W following surgery to induce lymphedema, we injected a 20 kDa PEG molecule conjugated to an 800IR CW NHS ester dye (LI-COR Biotechnology, Lincoln, NE, USA) intradermally into the mouse tail. Bestatin treatment started three days after surgery and continued daily until the 4W timepoint. As the active contractions from lymphatic pumping create intensity fluctuations, movement of these “packets” that are reflective of the vessel stroke volume can be quantified using live NIR imaging. In so doing, we found that lymphatic function as determined by measuring the lymphatic contractile transport of the tracer normalized to the presurgery baseline was significantly increased at 2W in bestatin-treated mice compared to the saline group (Figure 6A), an effect that was lost by 4W. Other metrics, including lymphatic contractile frequency, amplitude, and pumping pressure showed no significant change in bestatin-treated mice compared to the saline group (Appendix A). Even with this measured change in lymphatic contractile function, bestatin did not reduce tail swelling compared to the saline group at any of the timepoints analyzed (Figure 6B). To measure nanoparticle uptake to and accumulation within the dLNs, we harvested the dLNs at 1W, 2W, and 3W after single vessel ligation surgery. As described previously, one day prior to LN harvest, NIR-labeled NPs were injected intradermally into the tail. After LN harvest, we again used our NIR imaging setup to take fluorescent images of the dLNs and quantify maximum, normalized mean, and normalized sum of fluorescence (Figure 6C). We found that bestatin-treated mice had increased NP accumulation in the intact vessel dLN at 3W compared to the saline control group, quantified through both normalized mean fluorescence and normalized sum of fluorescence (Figure 6D). We also used multiple linear regression to analyze how both treatment and time correlated with NP fluorescent measurements. The coefficient of the interaction term between treatment and time in the linear regression model was significantly correlated for normalized sum of fluorescence (*p* = 0.0307) and close to significant for normalized mean fluorescence (*p* = 0.0509), showing that the slopes of these NP fluorescent measurements in the intact vessel dLN versus time were different between treatment groups. However, fluorescence in the injured vessel dLN at 3W showed no difference between the bestatin and saline groups (Appendix A). Additionally, there were no significant differences in the intact/injured ratio for these fluorescence metrics between the bestatin and saline groups (Figure 6E). These results suggest that LTB4 antagonism through bestatin treatment increases transport and accumulation of NPs to the intact vessel dLN during lymphedema progression. The return of fluid transport to the injured vessel dLN, however, was not significantly altered by bestatin treatment. Taken together, these results suggest that antagonism of LTB4 through bestatin treatment increases transport and accumulation of NPs to the dLNs, specifically in the vessel left intact following single vessel ligation surgery.

### 2.7. Bestatin Treatment Has No Significant Effect on Magnitude of Immune Response in dLNs during Lymphedema Progression

Given the observed increase in lymphatic function following bestatin treatment, we were interested in studying if LTB4 mediated the observed immune cell expansion in the dLNs in the context of lymphedema. Using the flow cytometry protocol described earlier, we found that bestatin treatment had no significant effect on either number of leukocytes or frequency of leukocytes in the summed dLNs for the leukocyte subsets analyzed compared to the saline control group (Figure 7A,B). This data suggests that LTB4 did not significantly modulate the increase in leukocytes in the dLNs observed during lymphedema progression. We also found that swelling correlated with both number and frequency of cells in the intact node for bestatin-treated mice similarly to the correlations observed in untreated mice (Appendix A). This result shows that in bestatin-treated mice the expansion of leukocytes in the dLNs was still directly correlated with the observed tail swelling.

### 2.8. Bestatin Treatment Leads to Leukocyte Expansion in the Intact Vessel dLN Compared to the Injured Vessel dLN

We also wanted to analyze how LTB4 antagonism would differentially affect immune cell expansion in the intact vessel dLN versus the injured vessel dLN. We again analyzed the LNs separately at each timepoint, comparing total cell number between the intact and injured nodes for bestatin-treated mice. For bestatin-treated mice, the intact vessel dLNs had more T and B cells, myeloid cells, dendritic cells, monocytes, and macrophages than the injured vessel dLNs at 2W (Appendix A). Normalized difference was also significantly higher in the bestatin group compared to the control group at 2W for these leukocyte subsets (Figure 8A). Taken with the results from Figure 6A showing an increase in lymphatic contractile function following bestatin treatment at 2W after surgery, this data suggests that this increased lymphatic function may have been partially responsible for the differential leukocyte expansion between nodes at 2W following bestatin treatment. We also plotted NP fluorescence metrics versus normalized difference to determine if bestatin treatment altered the correlations observed between lymphatic function and dLN leukocyte expansion in the untreated group. For the bestatin-treated group, the intact/injured ratios of maximum fluorescence and normalized mean fluorescence showed no correlation with the normalized difference of cells in the dLNs, while the intact/injured ratio of normalized sum of fluorescence did show a positive correlation with normalized difference (Figure 8B). The intact/injured ratio of lymph node area was still positively correlated with normalized difference (Figure 8B). These results show that there was still a correlation between NP fluorescence in the dLNs and leukocyte expansion following bestatin treatment; however, the effects of bestatin treatment on normalized difference at 2W likely explain the loss of this correlation for maximum fluorescence and normalized mean fluorescence.

## 3. Discussion

The immune response during lymphedema development has been previously well characterized in lymphedematous tissue [3,4,5,6,7,8,9,10,12,13]. This study provides the first evidence that total levels of leukocytes within LNs draining the diseased tissue bed significantly increase during lymphedema progression (Figure 1A). Whether this is the result of local recruitment versus proliferation, however, remains to be determined. The magnitude of lymphocyte expansion within the dLNs, which make up the majority of leukocytes within the lymph nodes, is not likely explained solely due to migration of lymphocytes into the lymph node as lymphocytes are not known to extensively traffic to lymph nodes during inflammation [30,31,32,33,34]. In fact, a previous study has suggested that CD4+ T cells migrate from dLNs to sites of injury during lymphedema development, although the mechanisms facilitating this migration remain unclear [13]. This efflux of T cells from the dLNs may counteract the expansion of LN resident T cells, possibly explaining why the frequency of T cells within the dLNs decreases during lymphedema progression, even as the overall number of T cells within the dLNs increases. Our data suggests that CD4+ T cells increase as a percentage of T cells within the dLNs during lymphedema progression. Interestingly, the mice with the most swollen tails at 2W following surgery also had the highest frequency of T helper cells within the dLNs (Figure 2C). Other studies which have analyzed T helper cell involvement in lymphedema have used the complete ligation mouse tail lymphedema model, which induces a larger swelling response than that seen in the single vessel ligation model used in our study [6,13]. Taken together, it seems likely that T helper cells make up most T cells within both the dLNs and lymphedematous skin during significant tissue swelling. Our results show that the previously identified T helper cell response is present in the context of lymphedema development when lymphatic drainage from the lymphedema site remains intact and that dLNs are likely critical in facilitating this immune response.

We have additionally identified that B cells increase in number during lymphedema development in the dLNs and overtake T cells as the predominant leukocyte subset within the dLNs (Figure 1A,B). Previous studies investigating B cell changes in lymphedematous skin found no differences in B cell populations between control and lymphedema groups [35]. Additionally, humoral immunity was shown to be impaired following immunization in a genetic mouse model that lacked dermal lymphatic drainage, suggesting that cell trafficking to lymph nodes along with passive antigen transport is required for strong humoral immunity [15]. The increase in B cells within the dLNs suggest that they are involved in the response as lymphedema develops; however, we did not directly measure antibody titers and cannot directly comment on the activity of these B cells. There is also evidence that B cells in inflamed lymph nodes may play a role in enhancing dendritic cell migration from distal tissue [36]. The role of humoral immunity and antibody production from B cells during lymphedema development has been understudied, and our data suggests that more research focused on elucidating the role of B cells in this process is needed.

Myeloid cells, primarily macrophages, have been suggested to play an important role in driving lymphatic failure and tissue fibrosis during lymphedema development [3,4,37]. Our data suggests that multiple myeloid cell subsets, including monocytes, macrophages, dendritic cells, and neutrophils increase in number within the dLNs during lymphedema development (Figure 1A). This increase is likely due to both migration of tissue-resident myeloid cells to the dLNs, and proliferation of LN resident myeloid cell populations [38,39]. Interestingly, normalized difference between the intact and injured lymph nodes was significantly increased at 1W compared to 3W for both monocytes and macrophages following lymphedema induction, while other leukocyte subsets did not show a significant increase (Figure 5A). Given that monocytes and macrophages are both migratory cell types, it is likely that their migratory capabilities are at least partially responsible for their observed increase in the dLNs. Future studies are needed to determine if migration of myeloid cells to the lymph nodes following lymphedema surgery initiates the observed expansion of leukocytes, or whether this growth is independent of myeloid cell migration.

Inflammation has been previously shown to induce lymphangiogenesis within the lymph node [36]. Our results suggest that the dLNs become much larger as lymphedema progresses and we have also identified that many leukocyte populations increase greatly in number in the dLNs. From our flow cytometry data, we also found that the number of cells not identified as leukocytes (CD3-, CD19-, CD11b-, CD11c-, Ly6C-) increased (data not shown). It is likely that these cells include stromal cells that form the structure of the lymph node and endothelial cells, including lymphatic endothelial cells. Thus, our data likely indicates that lymphedema progression induces lymphangiogenesis within the dLNs, contributing to the observed growth in lymph node size. Further study is needed to investigate if lymphangiogenesis within the dLNs may contribute to the increase in leukocyte populations.

One major question regarding the pathogenesis of lymphedema is whether the onset of swelling is due to a loss of lymphatic function, which initiates the resulting immune response, or whether this loss of lymphatic function is a byproduct of immune-mediated physiological changes. To begin to answer this question, we combined a detailed analysis of immune cell composition in the dLNs during lymphedema development with experiments designed to track nanoparticle transport and uptake to the dLNs. Our novel single vessel ligation lymphedema model additionally allowed us to determine how differences in transport to the dLNs may impact the resultant expansion of LN resident leukocytes. We found that transport to and accumulation of NPs within the intact vessel dLN as measured by the normalized sum of fluorescence did not change significantly during lymphedema progression, while NP accumulation within the injured vessel dLN correspondingly increased (Figure 4B,C). This explains why differences in NP transport between the dLNs at 2D and 1W were diminished at later timepoints (Figure 4D). Given that lymphatic transport in the injured collecting vessel is not restored over this time frame [19], this suggests the formation of collaterals downstream that facilitate flow to the injured vessel dLN. Interestingly, this collateralization is sufficient to support leukocyte expansion in the injured vessel dLN through either direct immune cell migration or proliferation within the dLN. By also analyzing leukocyte populations within these LNs at these timepoints, we found that this loss in differential transport was accompanied by a loss in differential leukocyte composition. Normalized difference, a metric to describe differential leukocyte composition between the dLNs, was significantly positively correlated with the ratio between the intact and injured vessel dLNs of fluorescence metrics measuring NP accumulation within the nodes (Figure 5B). Taken together, these results show that differences in transport to the LNs are correlated with differences in leukocyte expansion within the LNs in the context of lymphedema. This study is the first that we know of to show that leukocyte proliferation in the dLNs during lymphedema varies depending on the magnitude of transport into the dLNs. These differences may be due to increased trafficking of migratory leukocytes and direct drainage of inflammatory factors to the intact lymph node, driving proliferation of lymph node resident leukocytes. It is important to note that even though transport of NPs to the intact vessel dLN did not increase during lymphedema progression, previous studies have shown that magnitude and speed of immune cell migration to dLNs through lymphatics depends on factors other than bulk lymph flow [31]. It is likely that both increased cell migration and cell proliferation contribute to the observed increase in leukocyte expansion during lymphedema progression even without an increase in lymphatic drainage to the intact vessel dLN. Future work should more specifically identify the mechanisms driving the observed differential leukocyte proliferation.

To understand how anti-inflammatory treatment may affect the composition of leukocytes within the dLNs during lymphedema development, we treated mice with bestatin following surgery. Bestatin has previously been shown to ameliorate swelling through its pro-lymphangiogenic effects in the double vessel ligation mouse tail lymphedema while also partially rescuing lymphatic function [12]. Additionally, bestatin treatment in a contact dermatitis model showed potent anti-inflammatory effects, reducing skin infiltration of neutrophils and CD8+ T cells [40]. Surprisingly, we found that bestatin treatment did not reduce leukocyte expansion in the dLNs at any of the timepoints analyzed compared to a control group injected daily with saline (Figure 7A,B). Additionally, bestatin treatment did not resolve swelling at any of the timepoints (Figure 6B). These results differ from a previous report which had identified bestatin as a potential treatment for lymphedema [12]. Our single vessel ligation lymphedema model leaves a pair of intact collecting vessels on one side of the tail while the complete ligation model used in the previous study blocked all lymphatic flow from the tail. One explanation for the lack of improvement in swelling in our model is that the pro-lymphangiogenic mechanism of action for bestatin is most effective in preventing swelling in a model where all lymphatic flow is blocked and lymphangiogenesis is necessary to reconnect the lymphatic network along the tail. In the clinical context this would suggest that bestatin is perhaps most efficacious as a therapy in more advanced stages of lymphedema. In our model, the intact collecting vessel provides an existing drainage pathway and prevents complete fluid stagnation, so it is possible that LTB4 levels are not sufficiently high enough for antagonization of LTB4 to affect swelling. It is worth noting, however, that the severity of swelling in the single vessel model and the complete ligation model are not markedly different.

Ketoprofen, a non-steroidal anti-inflammatory drug acting on the metabolism of arachidonic acid upstream of LTB4, has been tested as a treatment for lymphedema in a clinical trial [10,41]. Ketoprofen was shown to reduce skin thickness and improve histopathological characteristics of lymphedema but did not reduce limb volume in these patients [41]. In the case of chronic lymphedema, a primary source of excess volume is likely adipose hypertrophy, which seems to respond less to anti-inflammatory treatment [41]. In this study, we did not analyze histopathological changes in tail tissue following bestatin treatment but the observed increase in lymphatic function demonstrates the therapeutic efficacy of bestatin, even though changes in limb volume were limited, similar to what has been observed thus far clinically.

It is also important to note that bestatin is used clinically to treat patients with chronic lymphedema, while our mouse model of lymphedema presents acute disease which eventually resolves. Chronic lymphedema is characterized by extensive and prolonged swelling, sustained tissue fibrosis and hardening, and adipose hypertrophy. These aspects of the disease are difficult to reproduce in animal models of acute disease, so our single vessel ligation model only recapitulates some aspects of the clinical manifestation of lymphedema. Potential benefits of bestatin on lymphatic function observed in this study may thus provide additional benefits in clinical lymphedema given that our model does not fully reproduce the chronic disease characteristics.

While bestatin treatment did not reduce swelling or inhibit leukocyte expansion in the dLNs in our lymphedema model, it did increase NP accumulation within the intact vessel dLN as lymphedema progressed (Figure 6D). Through analysis of both NP uptake to the dLNs and in vivo imaging of lymphatic collecting vessel pumping we show that bestatin treatment increases lymphatic function as lymphedema progresses, suggesting that LTB4 mediates the decrease in lymphatic function that occurs during lymphedema development (Figure 6A,D). Further study is needed to determine the mechanism through which LTB4 mediates these changes in lymphatic function, whether it is through direct action on collecting lymphatics or through activation of leukocytes and production of various cytokines known to regulate lymphatic contractility [42,43,44,45,46].

Interestingly, bestatin-treated mice showed significant differential leukocyte expansion between the dLNs at 2W compared to the saline control group (Figure 8A). Taken together with the NIR measurements of lymphatic function, this difference between the dLNs corresponded with increases in lymphatic contractile transport in bestatin-treated mice at 2W and in NP accumulation at 3W. It is likely that increased leukocyte migration and transport to the intact vessel dLN at 2W in bestatin-treated mice could explain the differential increase in leukocytes, but we did not directly measure cell migration in these experiments. It is important to note that the increase in lymphatic contractile transport was measured in the intact vessel and the increase in NP accumulation was measured in the intact vessel dLN. The intact vessel dLN will drive the immune response as it receives significantly more drainage from the injury site than the injured vessel dLN in the first week following surgery in this model. Additionally, the differential leukocyte expansion at 2W following bestatin treatment preceded the increase in NP accumulation at 3W in the intact vessel dLN. These results suggest that differences in lymph node leukocyte populations may precede changes in drainage to the lymph nodes. The immune response may thus drive some of the observed changes in lymphatic function in this model.

Through combining flow cytometry analysis of leukocyte populations in dLNs with novel NIR imaging techniques to quantify lymphatic transport, our work has begun to uncover the relationship between fluid uptake and immune cell involvement during lymphedema development. We show that differential NP transport into the dLNs during lymphedema development was correlated with differential leukocyte expansion in those dLNs (Figure 5B). Bestatin treatment increased NP transport to the intact vessel dLN in our model (Figure 6D). This increase in lymphatic function also extended the differential leukocyte development between the dLNs (Figure 8A). However, this improvement in functional lymphatic transport did not affect the swelling in these mice (Figure 6B). The growth of leukocyte populations in the dLNs was also not affected by the changes in lymphatic function (Figure 7A,B). These results suggest that improving transport from sites of lymphatic injury during lymphedema development is not sufficient to reverse the characteristic swelling or immune cell-mediated physiological changes. Additionally, differential leukocyte composition within the dLNs was measured at timepoints prior to later differences in transport to the intact vessel dLN, suggesting that changes in lymphatic function during lymphedema development may be mediated by the immune response rather than purely by changes in the mechanical properties of the collecting lymphatic vessels. Future treatments for lymphedema may need to target both lymphatic function and the immune system to fully reverse the swelling that characterizes the disease.

## 4. Materials and Methods

### 4.1. Surgical Lymphedema Model

We used the single vessel ligation model of lymphedema developed in our lab where a pair of collecting vessels on one side of the mouse tail were left intact while the remaining dermal initial lymphatics and collecting vessels were ligated and cauterized [19]. Eight-week-old male and female C57Bl/6 mice (Charles River Laboratories, Wilmington, MA, USA) were used for this study according to procedures approved by the Georgia Institute of Technology IACUC Review Board (Protocol # A100293 approved on 6 March 2019). All animals were first anesthetized using inhaled isoflurane (5% induction, 2% maintenance). All animals received incisions 1.6 cm from the base of the animal spanning 80–90% of the circumference of the tail with particular care to standardize the incisions as much as possible. All incisions were cauterized to prevent bleeding and fluid leakage. After the surgical procedures, both collecting vessels were checked with NIR imaging to ensure they were either severed or remained intact as appropriate. Previously, our group characterized the lymphatic physiology of the tail as having a dominant and a nondominant collecting vessel [28]. All surgeries done in this study involved ligation and cauterization of the dominant collecting vessel. In this manuscript, the lymph node downstream of the dominant collecting vessel is referred to as the injured vessel dLN while the lymph node downstream of the nondominant collecting vessel is referred to as the intact vessel dLN. Mice were euthanized at 2 days (*n* = 14), 1 week (*n* = 16), 2 weeks (*n* = 21), and 3 weeks (*n* = 15) after surgery.

### 4.2. Flow Cytometry

Both the sacral lymph nodes, which drain fluid from the tail, were harvested when mice were euthanized at 2 days, 1 week, 2 weeks, and 3 weeks after surgery. The lymph nodes were stored in a 1 mg/mL Collagenase D solution (Sigma-Aldrich, St. Louis, MO, USA) and incubated at 37 °C for 1 h for disassociation. We then pushed the lymph nodes through a 70 μm cell strainer to create a single cell suspension. A CD16/32 antibody blocking solution (Tonbo Biosciences, San Diego, CA, USA) was added for 5 min on ice. After spinning down for 5 min at 350× *g* and decanting the liquid, Zombie Green live/dead solution (BioLegend, San Diego, CA, USA) was added at room temperature for 30 min. Following a wash step, multiple antibodies conjugated to fluorophores were added to the suspensions for 30 min on ice. The antibodies used were APC anti-mouse F4/80 (1:40), APC/Cyanine7 anti-mouse Ly6G (1:100), Brilliant Violet 421 anti-mouse/human CD11b (1:33), Brilliant Violet 510 anti-mouse Ly6C (1:100), Brilliant Violet 605 anti-mouse CD3 (1:40), Brilliant Violet 711 anti-mouse CD64 (1:40), Brilliant Violet 785 anti-mouse CD19 (1:40), PE anti-mouse I-A/I-E (1:40), PE/Cy7 anti-mouse CD206 (MMR) (1:50), and PerCP anti-mouse CD11c (1:40) (BioLegend, San Diego, CA, USA). After labeling with antibodies, 2% PFA was added for 15 min on ice to fix the cells. The stained cells were then stored in FACS buffer (10 mg/mL BSA in PBS) for analysis. A BD Fortessa Flow Cytometer was used to run the samples. UltraComp eBeads (Thermo Fisher Scientific, Waltham, MA, USA) were used for single-stain compensation controls. BD FACSDiva was used to record results and FlowJo was used for all analysis of results. Our flow panel did not include a stain for CD45, so leukocytes were defined as CD3+/CD19+/CD11b+/CD11c+/Ly6C+. The gating strategy implemented in FlowJo is shown in Appendix A. Using these same methods, we analyzed T cell subpopulations using the following antibodies: APC anti-mouse CD4 (1:100) and PE anti-mouse CD8a (1:100) (BioLegend, San Diego, CA, USA). The gating strategy for the analysis of T cell subpopulations is shown in Appendix A.

### 4.3. Tail Swelling

Tail swelling was measured prior to surgery and when mice were euthanized. Tail images were taken using an iPhone camera with a ruler adjacent to the tail to provide a measurement reference and ImageJ was used to quantify swelling. The diameter of the tail was measured at 5–10 mm increments along the tail starting at the site of injury. The truncated cone method was used to determine the total volume of the tail given the diameters along its length. Percent volume change was measured by comparing the tail volume at endpoint to the volume prior to surgery.

### 4.4. Nanoparticle Synthesis and Characterization

SH-NP were synthesized as previously described [20,21,22,23,24]. Briefly, 500 mg of Pluronic F127 (Sigma-Aldrich, St. Louis, MO, USA) was dissolved in 10 mL of MilliQ water to form micelles. Under argon, 400 μL of propylene sulfide (TCI, Tokyo, Japan) was added and stirred for 30 m before the addition of 14 mg of thiolated initiator activated for 15 m in 322 μL of sodium methoxide (Sigma-Aldrich, St. Louis, MO, USA) [47]. After 15 m, 64 μL of 1,8-diazabicyclo[5.4.0] undec-7-ene was added, and the solution was capped and reacted for 24 h while stirring at 1500 rpm. The reaction was then uncapped and exposed to air for 2 h, then dialyzed in a 100 kDa membrane (Spectrum Labs, New Brunswick, NJ, USA) against 5 L of MilliQ water for 3 d with 6 water changes. NP were dye labeled by overnight reaction with an excess of IRdye 680RD maleimide (LI-COR, Lincoln, NE, USA) in 1X PBS, and excess dye was removed by size exclusion chromatography using Sepharose CL-6B resin (GE Healthcare, Chicago, IL, USA). NP were concentrated by spin filtration and sterile filtered through a 0.2 μm filter. NP diameter was measured using a Malvern ZetaSizer instrument. Characterization of these NPs is shown in Appendix A.

### 4.5. Nanoparticle NIR Functional Analysis

A total of 10 μL of 680-NPs were injected intradermally into the mouse tail tip while the mouse was under anesthesia 24 hrs prior to euthanasia. After euthanasia, the sacral lymph nodes draining the mouse tail were harvested and placed into a black 96-well plate. The lymph nodes were then imaged using a customized imaging system consisting of a Sutter Instrument Lambda LS Xenon arc lamp (Sutter Instrument, Novato, CA, USA), an Olympus MVX-ZB10 microscope (Olympus Life Science, Waltham, MA, USA), a 650 nm bandpass excitation filter (45 nm full-width half maximum, FWHM), a 720 nm bandpass emission filter (60 nm FWHM), and a 685 nm longpass dichroic mirror. Images were acquired with a Photometrics Evolve Delta 512 EM-CCD (Teledyne Photometrics, Tuscon, AZ, USA). The two nodes were imaged in the same field of view using 1x magnification and an exposure time of 50 ms.

Multiple metrics were used to quantify fluorescence in the lymph nodes. The maximum fluorescence of the lymph node was quantified as the maximum fluorescence value within the lymph node. The normalized mean fluorescence was quantified as the mean fluorescence within the lymph node subtracted by the mean background fluorescence of a window 50 pixels by 50 pixels located away from the lymph node. The normalized sum of fluorescence was calculated as the normalized mean fluorescence multiplied by the area of the lymph node in pixels.

### 4.6. Bestatin Treatment

Bestatin treatment started at three days after surgery, as a previous study showed that beneficial effects of bestatin treatment were dependent on starting the treatment at this time [12]. To make bestatin stock, bestatin (Cayman Chemical, Ann Arbor, MI, USA) was first dissolved in DMSO at a 4 mg/mL concentration. Aliquots of 100 μL of this solution were stored in −20 °C. Aliquots were diluted with 900 μL of sterile saline per aliquot prior to injection. This bestatin solution was then administered through daily intraperitoneal injection at a concentration of 2 mg/kg. The control group received daily intraperitoneal injections of equivalent volumes of sterile saline without DMSO. Bestatin-treated mice were euthanized at 1W (*n =* 15), 2W (*n =* 18), and 3W (*n =* 12) after surgery.

### 4.7. NIR Lymphatic Functional Analysis

NIR lymphatic imaging was performed according to previously published methods [19,27,28,29]. LI-COR IRDye 800CW NHS Ester (LI-COR Biosciences, Lincoln, NE, USA) was conjugated to 20 kDa methoxypolyethylene glycol amine (Simga-Aldrich, St. Louis, MO, USA) and stored as lyophilized aliquots at −20 °C. After reconstitution in saline, 10 μL of this tracer was injected intradermally into the tip of the tail for fluorescence imaging. The injection was given at an entry angle of approximately 10 degrees to an approximate depth of 1 mm to specifically target the lymphatic vasculature. Care was taken to position the injection as close to the midline of the tail as possible to avoid favoring one collecting vessel over the other. Images were taken with a customized imaging system consisting of a Sutter Instrument Lambda LS Xenon arc lamp (Sutter Instrument, Novato, CA, USA), an Olympus MVX-ZB10 microscope (Olympus Life Science, Waltham, MA, USA), a 769 nm bandpass excitation filter (49 nm full-width half maximum, FWHM), an 832 nm bandpass emission filter (45 nm FWHM), and an 801.5 nm longpass dichroic mirror. Images were acquired with a Photometrics Evolve Delta 512 EM-CCD (Teledyne Photometrics, Tuscon, AZ, USA). The field of view was centered on the mouse’s tail at the site of injury. This location ensured that both the intact collecting vessel and the injury site could be viewed. The small volume of fluid injection and the use of NIR to enhance tissue penetration ensured that only fluorescence in the deeper collecting lymphatics was visible downstream of the injection site. The animals were imaged continuously from the time of injection until 20 min post-injection with a 50 ms exposure time and a frame rate of 10 fps. Analysis of NIR functional metrics was performed during the steady-state period ranging from 5–20 min after injection, as defined previously [28]. Lymphatic contractile frequency, amplitude, transport, and pumping pressure were measured and recorded as previously published [19].

### 4.8. Statistical Analysis

For comparison of leukocyte populations in the draining lymph nodes among the various timepoints analyzed, Kruskal–Wallis tests with Dunn’s multiple comparisons tests were used. Spearman’s rank correlation was used to calculate correlations between swelling percentage and various leukocyte populations. Fluorescent measurements from the 680IR NP experiments were transformed using the natural logarithm to create a normally distributed dataset. Ordinary one-way ANOVA tests with Tukey’s multiple comparisons tests were then used to compare between timepoints. Kruskal–Wallis tests with Dunn’s multiple comparisons tests were used to compare the normalized difference of leukocyte populations in intact versus injured lymph nodes between timepoints. Linear regression was used to calculate correlations between normalized difference and fluorescent measurements. A mixed-effects model with Šídák’s multiple comparisons test was used to compare normalized lymphatic fluorescence transport from the NIR imaging measurements between the timepoints analyzed. Two-way ANOVA test with Šídák’s multiple comparisons tests were used to compare the natural logarithm of fluorescent measurements from the 680IR NP experiments between saline and bestatin-treated mice at each timepoint. The NP fluorescent measurements were transformed using the natural logarithm to create a normal distribution of the data. Multiple linear regression was also used to investigate how NP fluorescence metrics correlated with both treatment and timepoint. The multiple linear regression model used the natural logarithm of the NP fluorescence metric as the dependent variable, with treatment and timepoint as the independent variables. The least-squares regression included terms for both independent variables and the interaction between them. Multiple Mann–Whitney tests using the Holm–Šídák method were used to compare leukocyte populations in draining lymph nodes between saline and bestatin-treated mice at each timepoint. Multiple Mann–Whitney tests using the Holm–Šídák method were used to compare normalized difference between saline and bestatin-treated mice at each timepoint. GraphPad Prism was used for all statistical analysis.

## Figures and Tables

**Figure 1 ijms-22-04455-f001:**
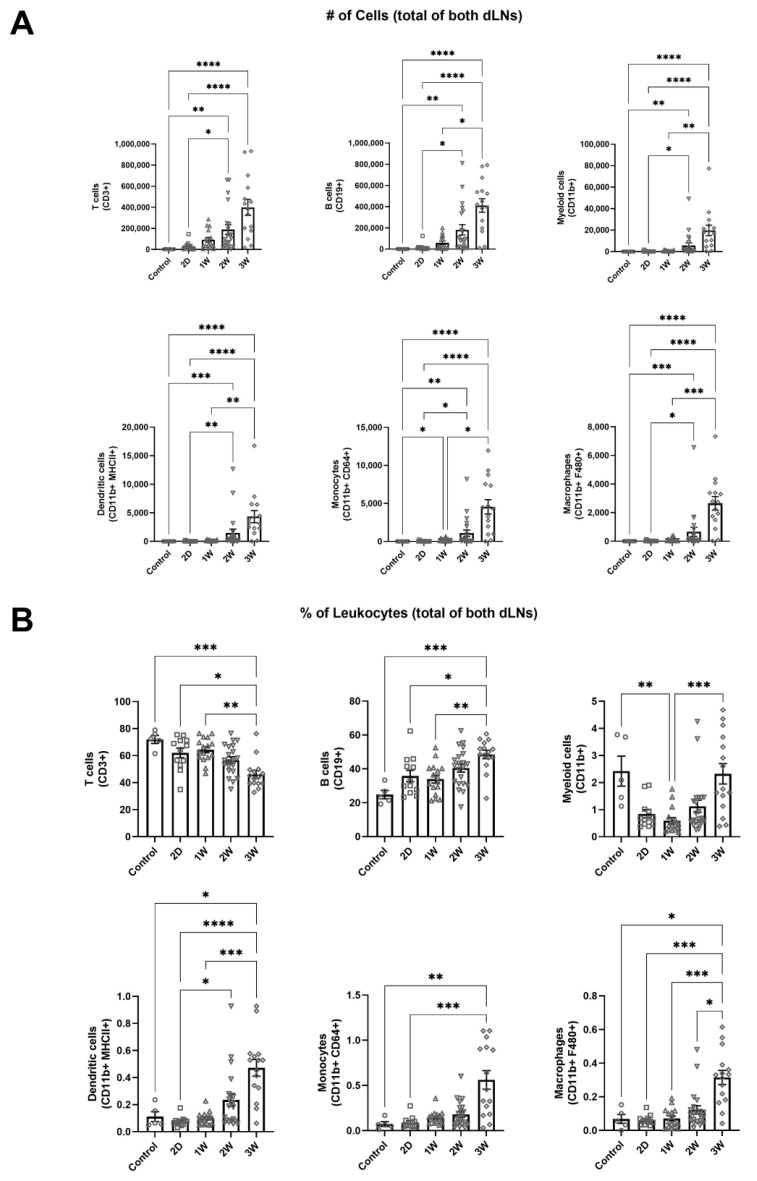
Leukocyte populations increase in dLNs during lymphedema progression. (**A**) Number of T cells, B cells, CD11b+ cells, dendritic cells, monocytes, and macrophages in dLNs for an unoperated control and after single vessel ligation lymphedema surgery at 2D, 1W, 2W, and 3W timepoints. Represented as the sum of the two dLNs. (**B**) T cell, B cell, CD11b+ cell, dendritic cell, monocyte, and macrophage frequencies of total leukocytes within dLNs for an unoperated control and after single vessel ligation lymphedema surgery at 2D, 1W, 2W, and 3W timepoints. Represented as the sum of the two dLNs for each leukocyte type divided by the sum of total leukocytes within the two dLNs. Kruskal–Wallis tests with Dunn’s multiple comparisons were used to compare between timepoints for both number of cells and percentage of leukocytes. Control (*n =* 6), 2D (*n =* 14), 1W (*n =* 16), 2W (*n =* 21), 3W (*n =* 15). Mean ± s.e.m. * (*p <* 0.05), ** (*p <* 0.01), *** (*p <* 0.001), **** (*p <* 0.0001).

**Figure 2 ijms-22-04455-f002:**
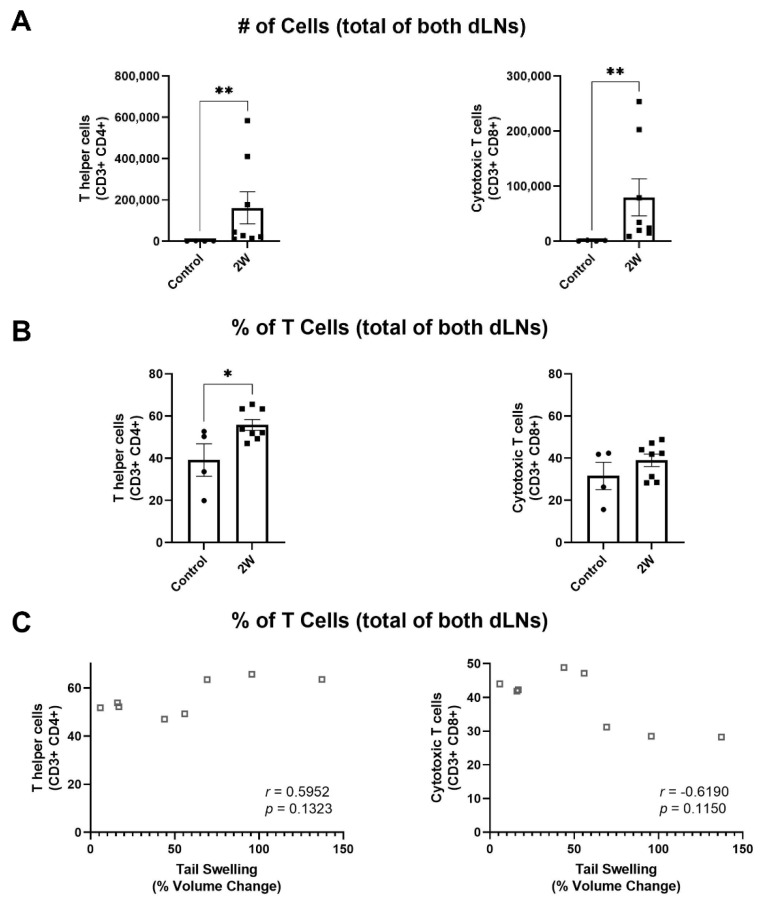
T helper cells increase as a percentage of T cells in dLNs during lymphedema development while cytotoxic T cells do not. (**A**) Number of T helper cells and cytotoxic T cells in dLNs for an unoperated control and after single vessel ligation lymphedema surgery at 2W timepoint. Represented as the sum of the two dLNs. Mann–Whitney tests were used for comparison between groups. (**B**) T helper cell and cytotoxic T cell percentages of T cells within dLNs for an unoperated control and after single vessel ligation lymphedema surgery at 2W timepoint. Represented as the sum of the two dLNs for T cell subtype divided by the sum of total T cells within the two dLNs. Unpaired T tests were used for comparison between groups. Control (*n =* 4), 2W (*n =* 8). Mean ± s.e.m. * (*p <* 0.05), ** (*p <* 0.01). (**C**) Spearman’s rank correlation between percent of T cells and percent volume change following single vessel ligation lymphedema surgery for T helper cells and cytotoxic T cells.

**Figure 3 ijms-22-04455-f003:**
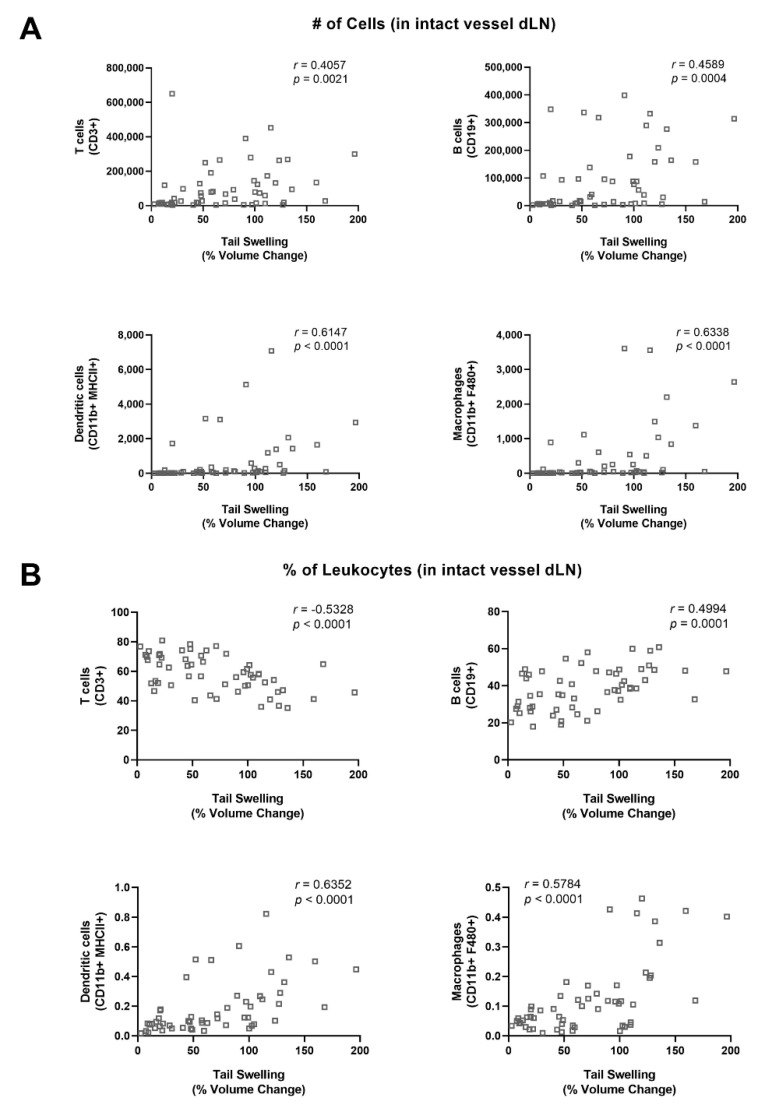
Leukocyte expansion in dLNs correlates with increase in tail volume after lymphedema surgery. (**A**) Spearman’s rank correlation between number of cells in intact vessel dLN and percent volume change following single vessel ligation lymphedema surgery for T cells, B cells, dendritic cells, and macrophages. (**B**) Spearman’s rank correlation between fraction of leukocytes in intact vessel dLN and percent volume change following single vessel ligation lymphedema surgery for T cells, B cells, dendritic cells, and macrophages.

**Figure 4 ijms-22-04455-f004:**
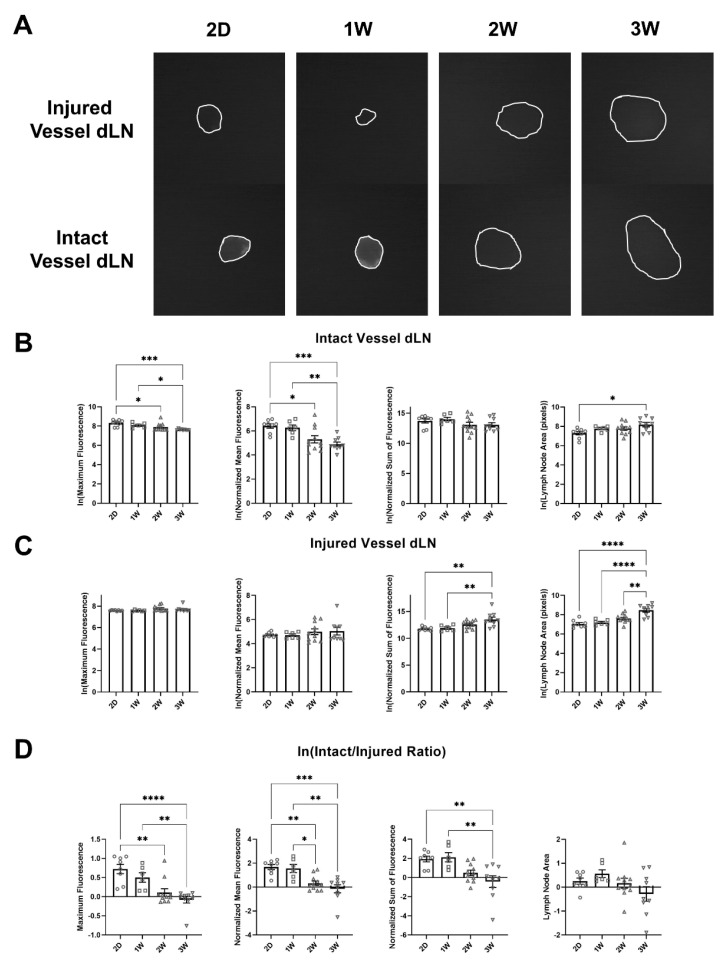
Lymph-draining nanoparticle accumulation within the intact vessel dLN decreases during lymphedema progression. (**A**) Fluorescent images of LNs harvested 24 h after injection of nanoparticles conjugated with 680 IR Dye at 2D, 1W, 2W, and 3W timepoints. (**B**) Natural logarithm of maximum fluorescence, normalized mean fluorescence, normalized sum of fluorescence, and lymph node area for the intact vessel dLN at 2D, 1W, 2W, and 3W timepoints. (**C**) Natural logarithm of maximum fluorescence, normalized mean fluorescence, normalized sum of fluorescence, and lymph node area for the injured vessel dLN at 2D, 1W, 2W, and 3W timepoints. (**D**) Natural logarithm of the intact/injured ratio for maximum fluorescence, normalized mean fluorescence, normalized sum of fluorescence, and lymph node area at 2D, 1W, 2W, and 3W timepoint. One-way ANOVA tests with Tukey’s multiple comparisons were used to compare between timepoints for B, C, and D. 2D (*n =* 8), 1W (*n =* 6), 2W (*n =* 11), 3W (*n =* 9). Mean ± s.e.m. * (*p <* 0.05), ** (*p <* 0.01), *** (*p <* 0.001), **** (*p <* 0.0001).

**Figure 5 ijms-22-04455-f005:**
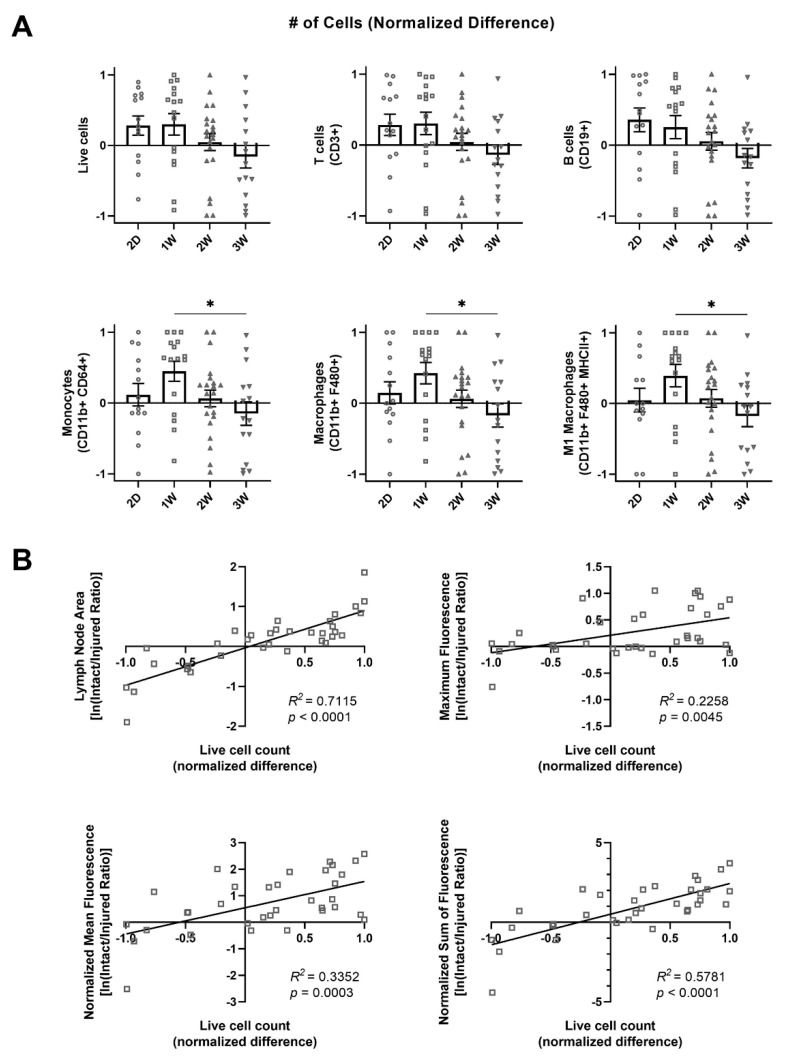
Differential leukocyte expansion between the dLNs is correlated with differential nanoparticle accumulation. (**A**) Normalized difference between intact and injured lymph nodes for number of live cells, T cells, B cells, monocytes, macrophages, and M1 macrophages at 2D, 1W, 2W, and 3W timepoints. Kruskal–Wallis tests with Dunn’s multiple comparisons were used to compare between timepoints. Mean ± s.e.m. * (*p* < 0.05). (**B**) Linear correlations and Pearson’s correlation coefficient (*R*^2^) between normalized difference of number of live cells and the natural logarithm of intact/injured ratio for lymph node area, maximum fluorescence, normalized mean fluorescence, and normalized sum of fluorescence.

**Figure 6 ijms-22-04455-f006:**
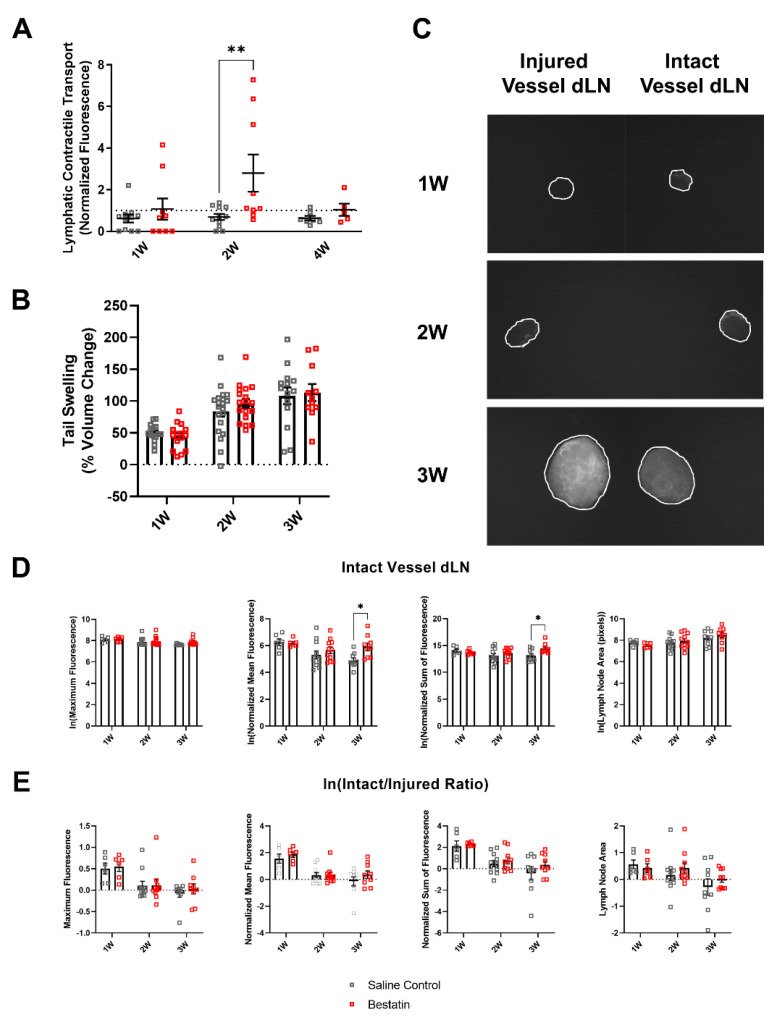
Bestatin treatment increases nanoparticle accumulation in the intact vessel dLN during lymphedema progression. (**A**) Lymphatic fluorescence transport measured from NIR imaging of in vivo lymphatic collecting vessel contraction at 1W, 2W, and 4W timepoints. A mixed-effects model with Šídák’s multiple comparisons was used to compare between treatments for each timepoint. Saline Control: 1W (*n =* 11), 2W (*n =* 11), 4W (*n =* 7); Bestatin: 1W (*n =* 9), 2W (*n =* 9), 4W (*n =* 5). * (*p <* 0.05), ** (*p <* 0.01). (**B**) Tail swelling following single vessel ligation surgery measured as percent change in volume from presurgery baseline for both bestatin-treated and saline control groups. Swelling of bestatin-treated mice was measured at endpoint, specifically at 1W, 2W, and 3W timepoints. (**C**) Fluorescent images of lymph nodes harvested from bestatin-treated mice 24 h after injection of nanoparticles conjugated with 680 IR Dye at 1W, 2W, and 3W timepoints. (**D**) Natural logarithm of maximum fluorescence, normalized mean fluorescence, normalized sum of fluorescence, and lymph node area in the intact node for the saline control group and the bestatin-treated group at 1W, 2W, and 3W timepoints. (**E**) Natural logarithm of the intact/injured ratio for maximum fluorescence, normalized mean fluorescence, normalized sum of fluorescence, and lymph node area for the saline control group and the bestatin-treated group at 1W, 2W, and 3W timepoint. Two-way ANOVA with Šídák’s multiple comparisons to compare between treatments at each timepoint for C and D. 1W (*n =* 6), 2W (*n =* 11), 3W (*n* = 9). Mean ± s.e.m. * (*p <* 0.05).

**Figure 7 ijms-22-04455-f007:**
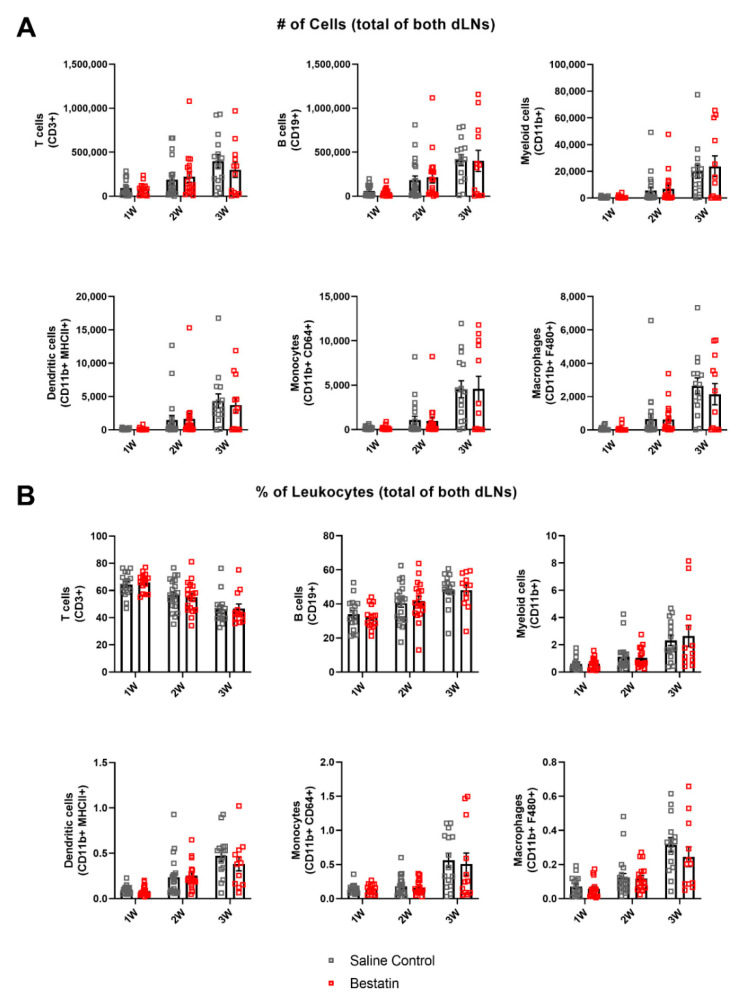
Bestatin treatment does not affect the overall leukocyte populations in the dLNs during lymphedema progression. (**A**) Number of T cells, B cells, CD11b+ cells, dendritic cells, monocytes, and macrophages in dLNs after single vessel ligation lymphedema surgery for the saline control group and the bestatin-treated group at 1W, 2W, and 3W timepoints. Represented as the sum within the two dLNs. (**B**) T cell, B cell, CD11b+ cell, dendritic cell, monocyte, and macrophage frequencies of total leukocytes within dLNs after single vessel ligation lymphedema surgery for the saline control group and the bestatin-treated group at 1W, 2W, and 3W timepoints. Represented as the sum within the two dLNs for each leukocyte type divided by the sum of overall leukocytes within the two dLNs. Multiple Mann–Whitney tests using the Holm–Šídák method were used to compare between treatments at each timepoint for both number of cells and percentage of leukocytes. 1W (*n =* 15), 2W (*n =* 18), 3W (*n =* 12). Mean ± s.e.m.

**Figure 8 ijms-22-04455-f008:**
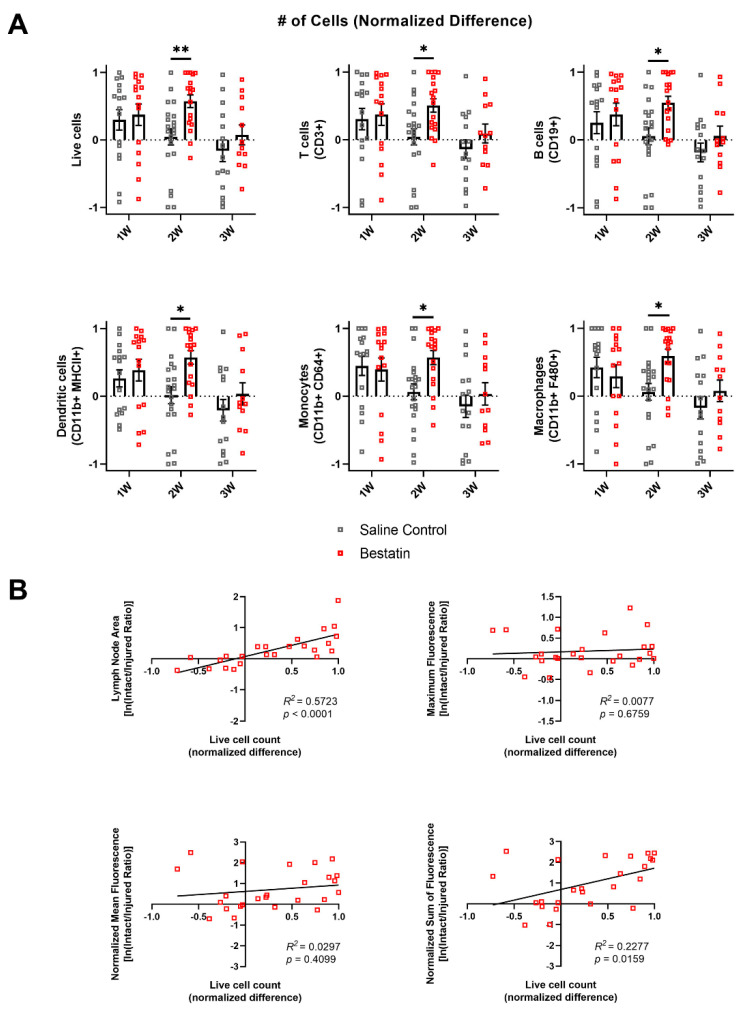
Bestatin treatment prolongs the differential leukocyte expansion of the intact compared to the injured vessel dLNs during lymphedema progression. (**A**) Normalized difference between intact and injured vessel dLNs number of total cells, T cells, B cells, dendritic cells, monocytes, and macrophages for the saline control group and the bestatin-treated group at 2D, 1W, 2W, and 3W timepoints. Multiple Mann–Whitney tests using the Holm–Šídák method were used to compare between treatments at each timepoint. Mean ± s.e.m. * (*p <* 0.05), ** (*p <* 0.01). (**B**) Linear correlations and Pearson’s correlation coefficient (*R*^2^) between normalized difference of number of live cells and the natural logarithm of intact/injured ratio for lymph node area, maximum fluorescence, normalized mean fluorescence, and normalized sum of fluorescence for the bestatin-treated group.

## Data Availability

The data presented in this study are available in this article and its accompanying Appendix A.

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
