# Peer review of "The Kinetics of Lymphatic Dysfunction and Leukocyte Expansion in the Draining Lymph Node during LTB4 Antagonism in a Mouse Model of Lymphedema"

_ijms, 2021, doi:10.3390/ijms22094455_

Round 1
Reviewer 1 Report
In this manuscript ‘The kinetics of lymphatic dysfunction and leukocyte expansion in the draining lymph node during LTB4 antagonism in a mouse model of lymphedema”, Cribb and colleagues employ an original mouse model to analyse leukocyte expansion in lymph nodes during lymphedema development. Interestingly, the authors used the mouse tail single vessel ligation model. This procedure leaves an intact lymphatic drainage pathway on one side of the tail while the other is blocked granting them the ability to compare leukocyte expansion in lymph nodes drained by either intact or injured lymphatic vessels.
This study thus sheds an unprecedented light on lymphedema-associated modifications occurring within lymph nodes and contributes to a better understanding of the immune response induced during lymphedema development.
The limitation of this manuscript is that having provided a detailed analysis of changes in lymph node leukocytes populations, the study failed to provide convincing data on how these modifications are induced and how they affect lymphedema progression and severity. The manuscript remains mostly at a descriptive level and there are significant gaps rendering the overall significance of content tricky to appreciate
Major comments:
The authors should go into more details regarding leukocytes subpopulations. For instance, T lymphocytes are only considered as CD3+ cells without any discrimination between CD4 and CD8 populations. Given the established contribution of CD4+ lymphocytes and especially Th2 type in lymphedema [Nores et al, Nat Commun, 2018. Zampell et al, Plos One, 2012] progression, it would be interesting to characterize the kinetics of CD4+ expansion in lymph nodes and whether there are significant changes in the balance between CD4+ and CD8+ within the T lymphocytes population during lymphedema progression.
In the same way data are lacking regarding M2 macrophage population, it would be interesting to know if M1/M2 populations are modified in lymph nodes as previously described in lymphedematous tissues [Ghanta et al,Am J Physiol Heart Circ Physiol. 2015].
The authors demonstrated an increase of leukocytes number in lymph nodes during lymphedema development but did not explain the origin of these newly formed cells. Is there an increase of cell proliferation inside the lymph nodes? Is there any lymphangiogenesis or any remodeling of the lymphatic vessel network inside the lymph nodes?
We can expect that this leukocyte expansion results from an increase of cell migration from the lymphedematous tissue to the draining lymph nodes. If it is the case, how can the authors explain the increase of leukocyte at 3W while there is a decrease of lymphatic drainage?
In the same way, what are the mechanisms supporting the leukocyte expansion in injured lymph nodes shown in Supplemental Figure 5 if there is no lymphatic drainage in the injured LN?
In Figure 3, fluorescent images are not convincing showing an increase of NP accumulation in 2W in intact LN while quantifications point to a decrease. Please change accordingly.
The authors should also provide data explaining the decrease in lymphatic drainage in intact LN since the lymphatic vessels draining it are still functional.
Reviewer 2 Report
The authors present a very well written paper, where they ask how acute lymphedema influences flow in functioning lymphatics and how this influences the immune cell content in the draining lymph node (dLN). They also ask how the anti-inflammatory bestatin influences this process in their lymphedema model. Overall the study is novel and the experiments and results are well described and controlled. Minor revisions to the figures would make it easier for the reviewer to follow the results and the discussion could be broadened.
Comments:
- The X-axis on the graphs are very hard to read and follow.
- In Figures 1, 2, 4, and 6, it would be better to label by cell type rather than antibodies used for FACS. Moreover, the figure legend and the x-axis labels are not consistent when discussing the cell types (antibody in x-axis, cell type in figure) making harder for the reader to follow.
- In the figures about the LN data, the bracketed labels in the graphs in the second line are common for all the graphs in a figure panel. This would be better presented as a title to the panel and the x-axis font could be increased to make it easier to read.
- Supplemental Figure 4 would be better presented in the main body of the paper in Figure 3. Maximum fluorescence, and normalized mean fluorescence should also be presented for the injured dLN as it is need to interpret Figure 3C ratio data.
- For the bestatin experiment, did control mice receive only sterile saline or sterile saline and equivalent amount of DMSO? The effect of DMSO in their model should be taken into consideration as it is 1/10 of the solution with bestatin.
- Bestatin is used in patients with chronic lymphedema and the presented model is an acute mouse model of lymphedema. The difference between the presented acute model vs. longer more chronic lymphedema should be addressed in the discussion.
- Inflammation is known to induce lymphangiogenesis in the lymph node. How does this correlate and contribute with your findings of flow and cellular content in the dLN?
Round 2
Reviewer 1 Report
By adding new experiments and performing several changes in the manuscript, the authors have successfully answered all my comments.